# FLOWBOT: Inducing LLM Workflows with Bilevel Optimization and Textual Gradients

Hongyeon Yu [1]  Young-Bum Kim [1]  Yoon Kim [2]

## Abstract

LLM workflows, which coordinate structured calls to individual LLMs/agents to achieve a particular goal, offer a promising path towards building powerful AI systems that can tackle diverse tasks. However, existing approaches for building such workflows generally rely on human-crafted pipelines and prompts, which presents a substantial bottleneck in real world deployment. How can we automatically induce LLM-based agents and workflows in a data-driven way? This paper describes a simple data-driven approach for automatically inducing agents and LLM workflows. We formulate workflow induction as a bilevel optimization problem: an outer loop which optimizes a high-level sketch of the workflow (in particular how the LLM calls should be structured), and an inner loop which optimizes each individual LLM call one-by one. Both loops are optimized with "textual gradients" where for the inner loop we optimize each component in a modular way through "backpropagating" textual gradients layer-by-layer. We find that LLM workflows discovered through our FLOWBOT (work**flow** induction through **b**ilevel **o**ptimization and **t**extual gradients) approach performs competitively against strong baselines that make use of human-crafted or generated workflows.

## 1. Introduction

Large language models (LLMs) are increasingly deployed not just as standalone models but as part of complex workflows: structured sequences of calls to LLMs, augmented with tools and intermediate reasoning steps, that collectively solve complex tasks. By orchestrating multiple LLM calls with different instructions and tool access, these "compound AI systems" (Zaharia et al., 2024) extend what a sin-

gle prompted LLM can do. However, existing approaches for building such workflows often rely on human-crafted pipelines, which can be hard to scale across tasks and domains.

How can we automatically induce and optimize LLM workflows in a data-driven way? Recent works have approached this problem from the perspective of prompt optimization (Khattab et al., 2024; Opsahl-Ong et al., 2024; Cheng et al., 2024; Agrawal et al., 2025; Yin & Wang, 2025), wherein individual components of an LLM workflow are optimized to improve the overall workflow performance. While various methods have explored for prompt optimization within this context, these works generally assume that the overall workflow structure is fixed. This work describes a simple approach for learning both the workflow structure as well as the individual components. We formulate LLM workflow induction as a bilevel optimization problem, where the outer loop optimizes a high-level *sketch* of the workflow— how LLM calls are structured and composed—while the inner loop optimizes each individual LLM call given a fixed sketch. Both levels are optimized with "textual gradients" (Pryzant et al., 2023; Yuksekgonul et al., 2025), i.e., natural language feedback generated by an LLM based on the input, output, and the loss. In the inner loop, we treat the workflow as a stack of modular layers and "backpropagate" textual gradients layer-by-layer, updating each component while conditioning on the behavior (and errors) of downstream components. In the outer loop, we use textual gradients to refine the workflow sketch itself, deleting existing calls, or adding new ones.

Our approach, dubbed FLOWBOT (work**flow** induction through **b**ilevel **o**ptimization and **t**extual gradients), brings ideas from neural architecture search (Zoph & Le, 2017; Liu et al., 2019; Pham et al., 2018; Elsken et al., 2019) into the space of LLM systems. The inner loop plays a role analogous to gradient-based parameter optimization within a fixed network architecture, while the outer loop mirrors architecture-level search over compositions of modules. Across multiple tasks, we find that workflows discovered by FLOWBOT perform competitively against strong baselines that optimize prompts within human-crafted workflows such as GEPA (Agrawal et al., 2025), as well as automated work-

[1]Naver Search US [2]Massachusetts Institute of Technology. Correspondence to: Hongyeon Yu <hongyeon.yu@navercorp.com>.

*Proceedings of the 43rd International Conference on Machine Learning*, Seoul, South Korea. PMLR 306, 2026. Copyright 2026 by the author(s).

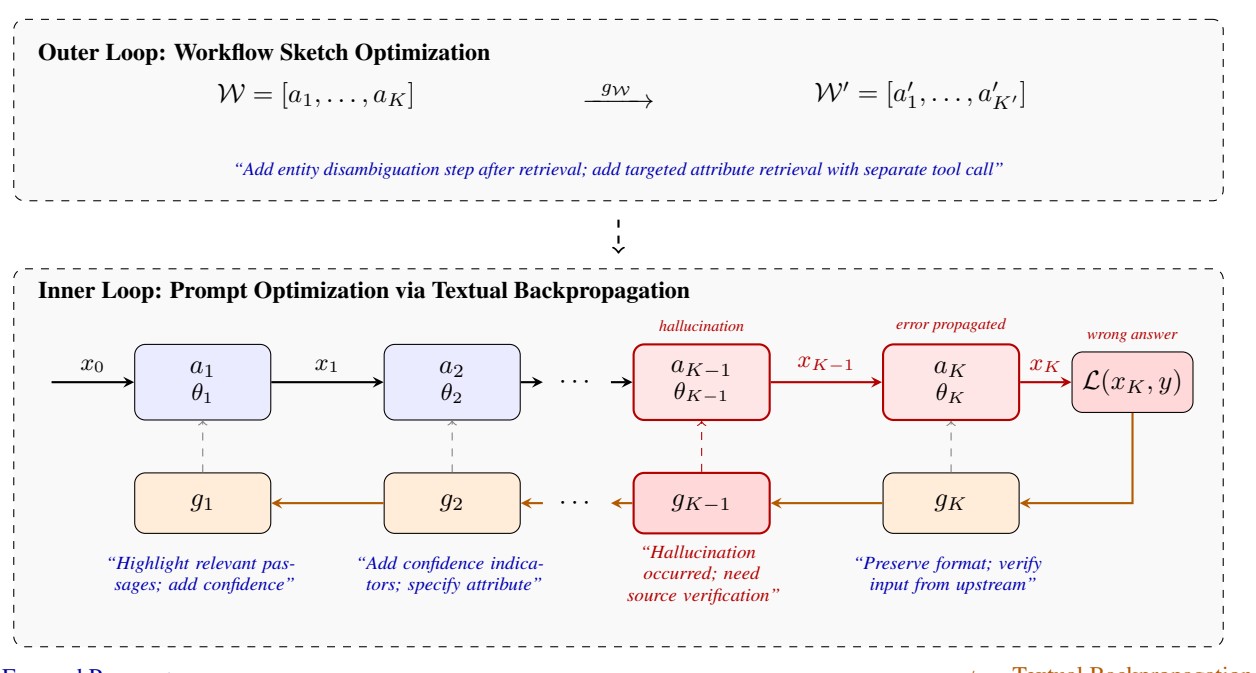

*Figure 1.* Overview of our FLOWBOT approach. The **outer loop** optimizes the workflow sketch $\mathcal{W}$ (the structure of LLM calls), while the **inner loop** optimizes each prompt $\theta_i$ via layer-wise textual backpropagation. Gradients $g_k$ are propagated backwards from the loss, analogous to backpropagation in neural networks. Example gradient snippets (in italics) are from a HotpotQA task where $a_{K-1}$ produced a hallucination that propagated to the final answer;.

flow generation approaches such as AFlow (Zhang et al., 2025). Our method induces sensible workflow decompositions and component behaviors, and ablations indicate that both the outer-loop structural search and the inner-loop modular optimization are both crucial for performance.

## 2. Methods

### 2.1. Problem Formulation

We define an LLM workflow as a chain $\mathcal{W} = [a_1, \dots, a_K]$ consisting of invocations to individual LLM calls $a_i$. The chain could consist of loops or conditional statements (e.g., $a_i$ could go back to $a_j$ for $j < i$). We parameterize $\mathcal{W}$ directly in text space, and thus each $a_i$ is a high-level description of the $i$-th LLM call (e.g., a retrieval step).[1] Each LLM call $a_i$ is associated with a detailed prompt $\theta_i$ that specifies its instruction, including the tools it has access to. The workflow $\mathcal{W}$ does not have access to the exact prompt $\theta_i$, but only the high-level description $a_i$. Letting $a_i(z) = \texttt{LLM}(z; \theta_i)$ be the output of applying $a_i$ with prompt $\theta_i$ to textual input $z$, the workflow $\mathcal{W}$ takes an input $x$ and produces an output via

$$f_{\mathcal{W}}(x) = (a_K \circ \cdots \circ a_1)(x).$$

Given a training set of input-output pairs $(x, y)$ from a distribution $\mathcal{D}$, our goal is to optimize the workflow to minimize

the loss $\mathcal{L}$ (e.g., negative of accuracy, or feedback from LLM-as-a-judge), i.e.,

$$\min_{\mathcal{W}, \{\theta_i\}_{i=1}^K} \mathbb{E}_{(x,y)\sim\mathcal{D}}[\mathcal{L}(f_{\mathcal{W}}(x), y)]$$

Making an analogy deep learning architectures and neural architecture search (Zoph & Le, 2017; Liu et al., 2019; Elsken et al., 2019), we can think of $\mathcal{W}$ as specifying the overall computation graph (e.g., number and type of layers, nonlinearities, etc.), while $a_i$ specifies the actual layer itself with optimizable parameters $\theta_i$.

### 2.2. Bilevel Optimization with Textual Gradients

We formulate the above as a bilevel optimization problem,

$$\min_{\mathcal{W}} \min_{\{\theta_i\}_{i=1}^K} \mathbb{E}_{(x,y)\sim\mathcal{D}}[\mathcal{L}(f_{\mathcal{W}}(x), y)],$$

where we optimize the inner and outer loops in turn. See Figure 1 for a high-level overview.

**Inner loop.** Given the workflow structure $\mathcal{W}$, we optimize the prompt $\theta_i$ associated with $a_i$ through backpropagating "textual gradients" (Yuksekgonul et al., 2025). Let $x_0$ be the input, and $x_i = \texttt{LLM}(x_{i-1}; \theta_i)$ be the output of the $i$-th "layer" (so we have $f_{\mathcal{W}}(x_0) = x_K$). Further suppose we have run a forward pass through the workflow to obtain $\{x_i\}_{i=1}^K$.

The final layer's textual gradient is given by another LLM with prompt $\theta_{\text{grad-loss}}$ that instructs it to provide feedback

---

[1]We freely go back and forth between the text representation of $a_i$ and its function representation, which takes in a text input $z$ and produces a text output $a_i(z)$.

and changes to $\theta_K$ given the output $x_K$, answer $y$, as well as the loss $\mathcal{L}(x_K, y)$:

$$g_K = \text{LLM}(x_K, y, \mathcal{L}(x_K, y); \theta_{\text{grad-loss}})$$

(See appendix A for the exact prompts used by all the LLMs.) For the intermediate layer, we backpropagate the gradients by similarly instructing an LLM with prompt $\theta_{\text{grad-call}}$ that instructs it to provide feedback and necessary changes to $a_k$ given the current input $x_k$, the output $x_{k+1}$, and the next layer's gradient $g_{k+1}$:

$$g_k = \text{LLM}(x_k, x_{k+1}, g_{k+1}; \theta_{\text{grad-call}})$$

Each gradient $g_k$ is a natural language description of how step $k$'s output should change to benefit downstream steps. Given a batch of $B$ samples $\{x^{(j)}, y^{(j)}\}_{j=1}^B$, we run the above procedure for each example in the batch to obtain textual gradients $\{g_i^{(j)}\}_{j=1}^B$ for all layers $i \in [K]$. We then perform an update for layer $i$ by instructing an LLM with prompt $\theta_{\text{optim-call}}$ that asks it to find a new prompt for $a_i$ given the current prompt $\theta_i$ and the gradients for a given batch $\{g_i^{(j)}\}_{j=1}^B$:

$$\theta_i \leftarrow \text{LLM}(\theta_i, \{g_i^{(j)}\}_{j=1}^B; \theta_{\text{optim-call}})$$

The above step results in an updated prompt $\{\theta_i\}_{i=1}^K$ for all LLM calls $\{a_i\}_{i=1}^K$. We also include the workflow sketch in the grad/optim prompts.

**Outer loop.** The outer loop optimizes the workflow sketch $\mathcal{W} = [a_1, \ldots, a_K]$, again with textual gradients. We feed the (text description of the) current workflow $\mathcal{W}$, along with the intermediate steps $\{x_i\}_{i=0}^K$, output $y$, and loss $\mathcal{L}(x_K, y)$, to obtain suggested changes to the current workflow,

$$g_{\mathcal{W}} = \text{LLM}(\mathcal{W}, \{x_i\}_{i=0}^K, y, \mathcal{L}(x_K, y); \theta_{\text{grad-workflow}}).$$

The suggested changes may include addition, removal, or merging of LLM calls. We obtain the workflow gradients for a batch of $B$ examples and update the workflow via an LLM with prompt $\theta_{\text{optim-workflow}}$

$$\mathcal{W} \leftarrow \text{LLM}(\mathcal{W}, \{g_{\mathcal{W}}^{(j)}\}_{j=1}^B; \theta_{\text{optim-workflow}}).$$

The inner/outer loops may be repeated multiple times for a given batch.

### 2.3. Initialization

Our optimization algorithm requires an initial workflow to optimize over iteratively. For initialization, we first give a batch of examples to an existing LLM and ask it to produce the output without any explicit instructions. We then perform a textual gradient step with respect to the workflow sketch (i.e., the outer loop) to obtain an initial sketch $[a_1, \ldots, a_K]$. Then for each $a_i$, we have another LLM generate an initial detailed instruction $\theta_i$ given the high-level description $a_i$. The full prompt for the initialization component is given in appendix A.3.

## 3. Empirical Study

We evaluate FLOWBOT across a diverse set of benchmarks to test whether fully automated workflow induction improves LLM workflows consistently across tasks.

### 3.1. Experimental Setup

We use the same setup for all datasets: one epoch of training with batch size 5, where for each batch we run two bilevel optimization loops, where one optimization loop itself consists of one outer loop step and 5 inner loop steps. We keep track of performance on the validation set after each batch and pick the best performing workflow on based on the validation set. We then run the best performing workflow on the test set.

**Baselines.** Our main prompt optimization baseline is GEPA (Agrawal et al., 2025), which performs evolutionary search to search for optimal prompts given a workflow structure. GEPA itself outperforms strong prompt optimization baselines such as Trace (Cheng et al., 2024), MIPROv2 (Opsahl-Ong et al., 2024), and TextGrad (Yuksekgonul et al., 2025); we include numbers from these other baselines for completeness. For baselines that perform automatic workflow generation, we mainly compare against AFlow (Zhang et al., 2025), which outperforms other workflow generation baselines such as ADAS (Hu et al., 2024). Both workflow approaches search over workflow structures described in program space (rather than text space, as in our work), and moreover do not employ "backpropagation"-based modular updates to individual components.

**Datasets.** We evaluate on ten benchmarks, a union of those used in the original GEPA/AFlow works: HotpotQA, IFBench, HoVer, PUPA, DROP, GSM8K, MATH, HumanEval and MBPP. These benchmarks span multi-hop QA, instruction following, multi-hop verification, privacy-conscious delegation, code generation, and math reasoning. HotpotQA is evaluated in two formats and we treat them as distinct benchmarks (Yang et al., 2018). HotpotQA1 is tool-augmented (question-only with retrieval), whereas HotpotQA2 is tool-free (question with provided context). We treat these separately since GEPA used the tool-augmented version, while AFlow did not. IFBench evaluates verifiable instruction following under explicit constraints (Pyatkin et al., 2025). HoVer evaluates many-hop fact extraction and claim verification over Wikipedia (Jiang et al., 2020). PUPA uses PAPILLON to study privacy-conscious delegation (Li et al., 2025). DROP is a reading comprehension benchmark requiring discrete reasoning over paragraphs (Dua et al., 2019). GSM8K and MATH evaluate mathematical reasoning with verifiable final answers (Cobbe et al., 2021; Hendrycks et al., 2021). HumanEval and MBPP evaluate code generation via unit tests (Chen et al., 2021; Austin et al., 2021). We follow the evaluation protocol and metrics as used in the GEPA and AFlow papers.

| Method | HotpotQA1 | IFBench | HoVer | PUPA | Avg. |
|---|---|---|---|---|---|
| Baseline | 38.00 | 47.79 | 46.33 | 78.57 | 52.67 |
| Trace (Cheng et al., 2024) | 60.33 | 51.19 | 46.00 | 74.18 | 57.93 |
| MIPROv2 (Opsahl-Ong et al., 2024) | 58.00 | 49.15 | 48.33 | 83.37 | 59.71 |
| TextGrad (Yuksekgonul et al., 2025) | 62.33 | 48.64 | 47.67 | 85.68 | 61.08 |
| GEPA (Agrawal et al., 2025) | 69.00 | 52.72 | 51.67 | 94.47 | 66.97 |
| GEPA+Merge (Agrawal et al., 2025) | 65.67 | **55.95** | 56.67 | **96.46** | 68.69 |
| FLOWBOT (ours) | **72.80** | 52.51 | **63.20** | 90.67 | **69.80** |

*Table 1.* Performance of FLOWBOT against prompt optimization methods that work within a fixed workflow. All methods use GPT-4.1 mini. Numbers from baselines are from Agrawal et al. (2025).

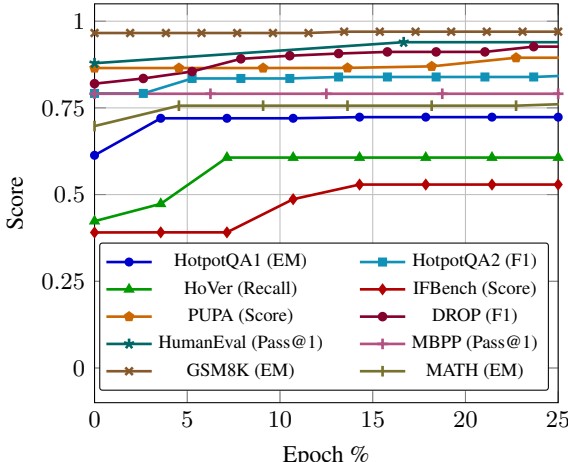

*Figure 2.* Validation performance of the best workflow seen so far across a single training epoch.

### 3.2. Main Results

**Learning.** To see whether our approach can learn from data, we show the "learning curve" of our method for all datasets in Figure 2, where we plot the performance of the best workflow seen so far (as measured by performance on the validation set). We indeed find that our approach can leverage data to find effective workflows. Our approach is moreover data-efficient, which is an advantage (less API costs) but also highlights the limitations—however, this kind of rapid ceiling hitting is often observed in prompt/workflow optimization approaches (Agrawal et al., 2025).

**Comparison against prompt optimization methods.** We compare against GEPA and same datasets used in the original work: HotpotQA1, IFBench, HoVer, and PUPA. GEPA evolves prompts through reflective edits guided by rollout-based feedback while GEPA+Merge further merges complementary prompt updates across candidates. For a controlled comparison, all methods (including ours) use GPT-4.1 mini for all LLMs. Table 1 shows that our method improves upon GEPA on the retrieval/verification-heavy benchmarks, while remaining competitive on instruction- and privacy-centric tasks. This is despite the fact that baselines start from hand-optimized workflows and prompts, while FLOWBOT has to discover the workflow from scratch.

| Benchmark | API Calls | | Cost | |
|---|---|---|---|---|
| | AFlow | FLOWBOT | AFlow | FLOWBOT |
| HotpotQA | 187K | 18.8K | $87.26 | $22.19 |
| DROP | 132K | 14.0K | $24.46 | $6.97 |
| HumanEval | 5.8K | 505 | $1.49 | $0.57 |
| MBPP | 430 | 2.7K | $0.05 | $2.99 |
| GSM8K | 1.3K | 14.3K | $0.42 | $7.68 |
| MATH | 11.9K | 4.5K | $17.28 | $7.26 |
| Total | 338K | 54.7K | $130.95 | $47.67 |

*Table 2.* Comparison of number of API calls and cost until the best validation performance is reached.

**Comparison against automatic workflow generation methods.** We follow prior end-to-end workflow optimization work and evaluate on the same six benchmarks used in AFlow: HotpotQA2, DROP, HumanEval, MBPP, GSM8K, and MATH. The original AFlow (Zhang et al., 2025) work used Claude-3.5 Sonnet as their meta LLM optimizer and GPT-4o mini as the executor LLM. For fair comparison, we reimplement AFlow using their open-source code base with GPT-4.1 mini. The results are shown in Table 3. Our method outperforms AFlow on four of the six benchmarks, indicating that bilevel workflow induction yields consistent improvements against AFlow when compared under the same model capacity.

**Cost.** How costly is our approach compared to other workflow induction approaches such as AFlow? For both AFlow and FLOWBOT, we compare the number of API calls and the cost until the best validation performance is reached for each task.[2] We use GPT-4.1 mini for both approaches. The results are shown in Table 2. We find that FLOWBOT generally requires fewer API calls and less cost.

### 3.3. Qualitative Analysis

**Analysis of induced workflows.** We examine the workflow structures discovered by our method and compare them against human-designed workflows from GEPA. Table 4 shows the induced workflows. Our bilevel optimization

---

[2]Running for full iterations for both approaches would result in 803K calls and $352.46 for AFlow, and 145K calls $165.39 for FLOWBOT. The cost is as of April 2026.

| Method | HotPotQA2 | DROP | HumanEval | MBPP | GSM8K | MATH | Avg. |
|---|---|---|---|---|---|---|---|
| *Prompt-based methods (GPT-4o mini)* | | | | | | | |
| Direct IO prompting | 68.10 | 68.30 | 87.00 | 71.80 | 92.70 | 48.60 | 72.75 |
| Chain-of-Thought (CoT) | 67.90 | 78.50 | 88.60 | 71.80 | 92.40 | 48.80 | 74.67 |
| CoT + Self-Consistency | 68.90 | 78.80 | 91.60 | 73.60 | 92.70 | 50.40 | 76.00 |
| MedPrompt (Nori et al., 2023) | 68.30 | 78.00 | 91.60 | 73.60 | 90.00 | 50.00 | 75.25 |
| MultiPersona (Wang et al., 2024) | 69.20 | 74.40 | 89.30 | 73.60 | 92.80 | 50.80 | 75.02 |
| Self Refine (Madaan et al., 2023) | 60.80 | 70.20 | 87.80 | 69.80 | 89.60 | 46.10 | 70.72 |
| *Workflow induction methods (Claude-3.5 Sonnet optimizer + GPT-4o mini executor)* | | | | | | | |
| ADAS (Hu et al., 2024) | 64.50 | 76.60 | 82.40 | 53.40 | 90.80 | 35.40 | 67.18 |
| AFlow (Zhang et al., 2025) | 73.50 | 80.60 | 94.70 | 83.40 | 93.50 | 56.20 | 80.32 |
| *GPT-4.1 mini for optimizer and executor* | | | | | | | |
| AFlow (our reimplementation) | 75.41 | 79.69 | **94.65** | **74.78** | 86.16 | 72.01 | 80.45 |
| FLOWBOT (ours) | **80.19** | **92.28** | 93.74 | 73.84 | **96.01** | **72.71** | **84.80** |

*Table 3.* Comparison of FLOWBOT against automatic workflow generation methods and prompting baselines. All numbers except the last two rows are from Zhang et al. (2025).

| Benchmark | Method | Workflow Structure | Steps |
|---|---|---|---|
| HotpotQA | GEPA | Retrieve passages (T) → Summarize passages → Generate hop-2 query → Retrieve passages (T) → Summarize with context → Generate final answer | 6 |
| | Ours | Initial broad retrieval (T) → Entity extraction and query refinement → Focused retrieval (T) → Answer verification and validation → Generate final answer | 5 |
| HoVer | GEPA | Retrieve passages (T) → Summarize passages → Generate hop-2 query → Retrieve passages (T) → Summarize with context → Generate hop-3 query → Retrieve passages (T) | 7 |
| | Ours | Entity extraction → Search query generation → Initial retrieval (T) → Document verification and filtering → Iterative gap-filling retrieval (T) → Aggregate final evidence | 6 |
| IFBench | GEPA | Generate response → Self-correct response | 2 |
| | Ours | Constraint extraction → Conditional verbatim repetition → Answer content generation → Compliance verification → Feedback-based refinement → Format normalization | 6 |
| PUPA | GEPA | Craft redacted request → External LLM generation → Synthesize final response | 3 |
| | Ours | Requirement extraction → Targeted content generation → Verification and refinement → Privacy-preserving synthesis | 4 |

*Table 4.* Comparison against human-designed workflows from GEPA. Steps marked with (T) use external tools.

discovers workflow structures that are qualitatively similar to those designed by humans. For example, for HotpotQA both workflows follow a multi-hop retrieval pattern with intermediate processing steps. The induced workflow includes explicit entity disambiguation and query planning, which the human design handles implicitly through summarization. We show examples of the individual LLM calls and prompts in Table 5 for HotpotQA. The induced prompts are sensible, and when tools are available, the workflow is able to make appropriate use of tools.

**Analysis of textual backpropagation.**   Is our layer-wise backpropagation effective? Performing a systematic analysis here is difficult here since we cannot manually verify each textual gradient. We did however generally find them to be sensible, and go through an example here during a training run of HotpotQA1. In this example the model is

given the following multi-hop question: "The 1905-1904 college basketball team lead by G.W. Leach represented a university founded in what year?" The ground truth answer is 1846 (Bucknell University), but the workflow produced 1865. See Figure 3 for an overview.

*Forward pass.* The workflow executed six steps. Step 1 (Wikipedia Search) retrieved relevant passages including information about the Bucknell Bison basketball team with G.W. Leach as captain. Step 2 (Entity Disambiguation) correctly identified "Bucknell University" as the primary entity. Step 3 (Targeted Retrieval) successfully retrieved passages stating that Bucknell was founded in 1846. Step 4 (Answer Extraction) correctly extracted "1846" from these passages. However, Step 5 (Verification) introduced an error: despite receiving the correct answer "1846" from upstream, the LLM hallucinated that "G.W. Leach is historically known

| **Step 3: Focused retrieval** ($a_3$) | |
|---|---|
| **Type** **Tools** **Description** | ToolExecutor `wikipedia_search_topk` Perform focused retrieval using the refined queries generated in Step 2 to obtain precise, authoritative passages about key entities and critical attributes relevant to the question. |
| **Prompt** ($\theta_3$) | You are given one or more key entities extracted from a user's question, along with any critical quantitative, temporal, or contextual attributes relevant to the question. Your task is to perform focused retrievals to find precise, authoritative passages specifically about these entities and their key attributes. The goal is to gather comprehensive, relevant, and temporally contextualized information that supports accurately answering the original question, including clear evidence, explicit notes about the presence or absence of relevant information, and concise candidate extractions to facilitate downstream answer verification and consolidation. [...] |
| **Step 4: Answer verification and validation** ($a_4$) | |
| **Type** **Tools** **Description** | LLMExecutor None Verify, normalize, and rigorously validate candidate answers by cross-checking all evidence from initial and focused retrievals, applying confidence scoring, resolving ambiguities, and triggering iterative focused retrieval if evidence is insufficient or conflicting. |
| **Prompt** ($\theta_4$) | You are an expert reasoning agent responsible for verifying, validating, and selecting the most accurate candidate answer(s) to a given question by rigorously cross-checking all provided evidence. Your goal is to ensure that final answers are well-supported, strictly grounded in the retrieved evidence, normalized to expected formats, and accompanied by transparent justifications and confidence assessments. Avoid assumptions or relying on external knowledge unless explicitly indicated as authoritative and supplementary. [...] |

*Table 5.* Example of induced LLM call specifications from the HotpotQA workflow. Step 3 ($a_3$) uses tools for focused retrieval, while Step 4 ($a_4$) performs LLM-based answer verification.

as a coach of the Kentucky Wildcats," leading it to reject the correct answer and substitute Kentucky's founding year (1865). This error propagated to Step 6 (Final Answer), which output the incorrect "1865."

*Textual backpropagation.* As shown in Figure 3, the gradient $g_6$ for the final step noted that the output format was correct but the answer was wrong, attributing the error to upstream steps. The gradient $g_5$ for the verification step pinpointed the root cause: "The main issue is the inaccuracy of the foundational data: the step concludes that 1865 is the correct founding year, but the ground truth is 1846. This step needs to incorporate more robust verification mechanisms, such as cross-checking multiple authoritative sources [...]" The gradient $g_4$ for the extraction step confirmed that it "correctly extracts and outputs '1846'," but noted that the next step "identifies '1846' as inconsistent with authoritative historical data"—correctly diagnosing that the error originated downstream of extraction. Gradients for Steps 1–3 similarly confirmed correct operation. See Appendix B for the full gradient.

*Prompt update.* Based on the gradient feedback, the verification step's prompt was updated to address the identified hallucination issue. The key changes included: (1) explicit instructions to check for discrepancies among candidate answers, (2) a requirement to cross-reference with common knowledge or authoritative sources, (3) guidance to flag uncertainties and retain multiple plausible candidates with confidence scores rather than prematurely committing to a single answer, and (4) clarification that if uncertainty

| Method | HotpotQA1 | HotpotQA2 |
|---|---|---|
| FLOWBOT (ours) | **72.80** | **80.19** |
| *prompt optimization only* | 60.07 | 72.06 |
| *w/o bilevel optim* | 61.73 | 79.26 |
| *w/o layerwise backprop* | 71.13 | 76.51 |

*Table 6.* Ablation study on algorithmic components

remains high, the agent should indicate that no reliable answer could be determined. The complete before-and-after prompts are provided in Appendix B.5.

### 3.4. Ablations

We perform a series of ablations on HotPotQA1 and HotPotQA2, where we average across 5 different runs for each ablation study.

**Ablations on algorithmic components.** FLOWBOT formulates workflow induction as a bilevel optimization problem and performs layer-by-layer backpropagation of textual gradients.[3] How important are these components?

We first show a simple baseline that does just prompt optimization based on the initial workflow from the first batch (Table 6: *prompt optimization only*), and find that this does not perform well. We next try removing the bilevel optimization formulation, treating the entire workflow induction as a single layer that is optimized with textual gradients.

---

[3]While this layer-by-layer backpropagation approach was mentioned in Yuksekgonul et al. (2025), all their experiments were in the "single layer" case to the best of our knowledge.

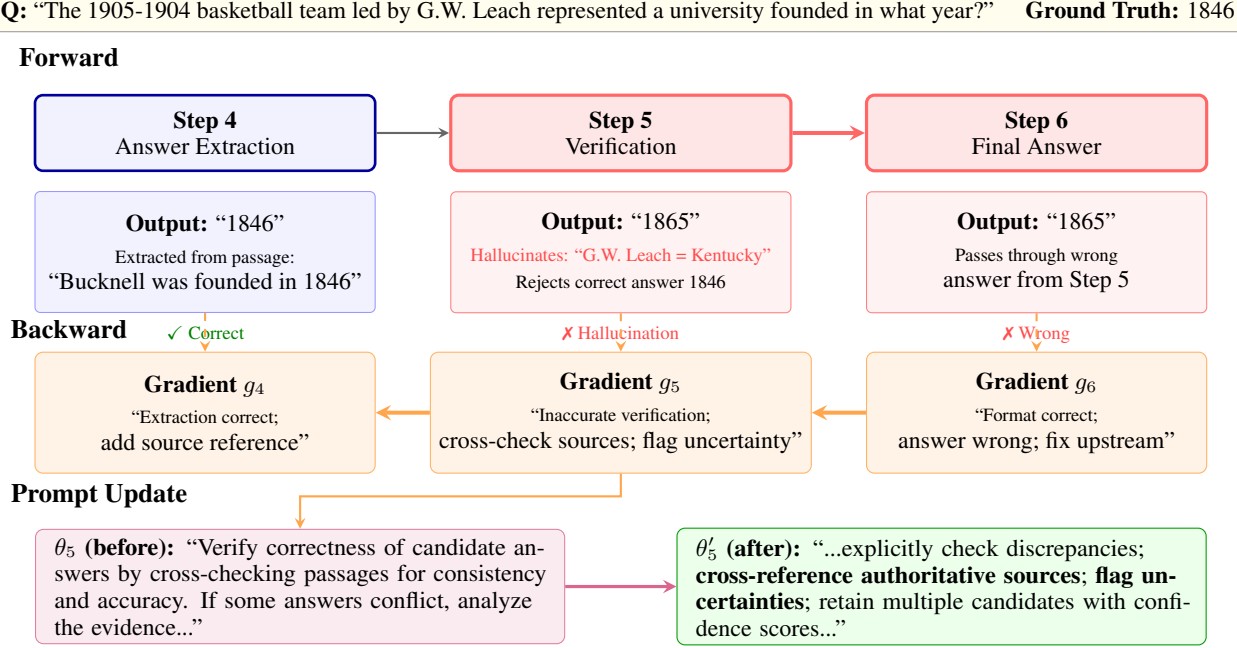

*Figure 3.* Textual backpropagation example on HotpotQA during training. **Forward (top):** Step 4 correctly extracts "1846" (Bucknell's founding year), but Step 5 hallucinates by incorrectly associating G.W. Leach with Kentucky (founded 1865) instead of Bucknell, rejecting the correct answer. **Backward (middle):** Textual gradients $g_6 \rightarrow g_5 \rightarrow g_4$ propagate error attribution via chain rule. **Prompt Update (bottom):** Gradient $g_5$ updates the verification prompt $\theta_5$ to cross-reference sources.

| Model | HotpotQA1 | HotpotQA2 |
|---|---|---|
| Qwen3.5-397B-A17B | 77.26 | 81.12 |
| GPT-OSS-120B | 69.46 | 80.61 |
| GPT-OSS-20B | 54.06 | 77.16 |

*Table 7.* Performance across models on HotpotQA1 and HotpotQA2 with different LLMs.

This underperforms the full approach (Table 6: *w/o bilevel optim*). We finally try keeping the bilevel optimization formulation, but performing the inner update in one go (Table 6: *layerwise backprop*). That is, instead of backpropagating layer by layer we ask an LLM to compute gradients for all steps $\{g_i\}_{i=1}^K$ in one go (and similarly ask another LLM to update the prompts $\{\theta_i\}_{i=1}^K$ in one go). We find that this approach results in substantial degradations. Collectively, these results indicate both bilevel optimization formulation and layer-by-layer textual gradient backpropagation are crucial for performance.

**Ablations with different LLMs.** We next test our approach with different open-weight LLMs. The results are shown in Table 7. We find that FLOWBOT works well with other open-weight models, with the Qwen3.5-397B model outperforming GPT-4.1 mini.

How does the performance vary as a function of the actual executor LLM and the meta LLM that produces the gradients and the prompt updates? Table 8 shows the results.

We find that strong meta LLMs enable significant improvements upon prompt-only optimization, which can enable significant improvements with smaller Executor LLMs. The improvements diminish with a larger executor LLM (i.e., Qwen3.5-397B), which makes sense since the meta LLM's job is more "difficult". Since the actual deployment will only involve the executor LLM, only requiring powerful LLMs for the meta LLM component is a practical aspect of our approach.

### 3.5. Extension: Inducing a Unified Workflow

We conclude with a proof-of-concept experiment that highlights the flexibility of FLOWBOT. Here, instead of having a separate workflow for each task, we induce a *single* unified workflow for all tasks. This requires the workflow to have *input-dependent* workflows where (for example) certain steps are skipped if it is not relevant for the task. We show the results in Table 9 with Qwen3.5-397B-A17B as both the executor and the optimizer. While the unified workflow predictably underperforms the task-specific workflows, we find that the unified approach performs quite well for many tasks, as shown in Table 10. The workflow is able to deduce that the 6 tasks can be split into 4 meta tasks: reading comprehension, math, code generation, and general conversion. It then uses the first step to route the question to these meta tasks. While far from perfect, we see this as a promising extension that could pave the path towards being able to induce more general-purpose workflows.

| Executor LLM | HotPotQA1 | | | HotPotQA2 | | |
|---|---|---|---|---|---|---|
| | No Opt | Prompt Opt | FLOWBOT | No Opt | Prompt Opt | FLOWBOT |
| Qwen3.5-4B | 0.417 | 0.513 | 0.681 | 0.608 | 0.615 | 0.761 |
| Qwen3.5-9B | 0.497 | 0.615 | 0.737 | 0.677 | 0.700 | 0.772 |
| Qwen3.5-27B | 0.618 | 0.677 | 0.779 | 0.774 | 0.775 | 0.800 |
| Qwen3.5-122B-A10B | 0.670 | 0.667 | 0.746 | 0.771 | 0.730 | 0.817 |
| Qwen3.5-397B-A17B | 0.682 | 0.732 | 0.740 | 0.784 | 0.816 | 0.806 |

*Table 8.* Performance on HotPotQA1 and HotPotQA2 across executor models, where the meta LLM for generating textual gradients and prompt updates is fixed to be Qwen3.5-122B-A10B. No Opt is the result of ordinary prompting. Prompt Opt optimizes the prompt only based on the initial workflow.

| Benchmark | Separate Workflows | Unified Workflow |
|---|---|---|
| HotpotQA2 | 81.12 | 79.20 |
| DROP | 92.14 | 92.00 |
| HumanEval | 97.40 | 84.40 |
| MBPP | 78.89 | 63.60 |
| GSM8K | 97.27 | 97.00 |
| MATH | 80.25 | 66.70 |
| **Average** | **87.85** | **80.48** |

*Table 9.* Performance comparison between inducing separate workflows and a single unified workflow.

| Step | Type | Description |
|---|---|---|
| 0 | Cond. / LLM | Classify task type and route |
| | | `READ_COMP` $\rightarrow$ 1–3 |
| | | `MATH` $\rightarrow$ 4–6 |
| | | `CODE_GEN` $\rightarrow$ 7–10 |
| | | `GENERAL` $\rightarrow$ 10 |
| | *Reading Comprehension* | |
| 1 | LLM | Evidence extraction |
| 2 | Loop / LLM | Answer + verification loop |
| 3 | LLM | QA output |
| | *Math Problem* | |
| 4 | LLM | Problem analysis |
| 5 | Loop / LLM | Solve + verification loop |
| 6 | LLM | Math output |
| | *Code Generation* | |
| 7 | LLM | Requirements analysis |
| 8 | LLM | Code generation |
| 9 | Loop / LLM | Code verification loop |
| 10 | LLM | Code output |
| | *General Conversation* | |
| 10 | LLM | General output |

*Table 10.* Induced input-dependent conditional workflow for AFlow benchmarks.

## 4. Discussion & Limitations

Our work has several limitations. For one, the optimization procedure incurs nontrivial computational overhead, since both structure and prompts are refined using LLM calls. This cost is particularly salient for tasks where minimal workflows already perform well, and where additional steps may introduce avoidable latency and complexity. More generally, the framework trades optimization-time compute for reduced reliance on manual workflow engineering. While our approach is data-efficient, in some cases the model converges to the final workflow and does not improve with more training data. This type of rapid convergence is often seen in such prompt/workflow optimization approaches. This suggests that more flexible learning mechanisms could enable workflow induction algorithms that can better leverage data.

## 5. Related Work

**LLM workflows and agentic systems.** LLMs are increasingly deployed inside structured workflows that combine multiple model calls, tools, and retrieval steps rather than as single-shot predictors. Early work on tool-augmented LLMs such as ReAct (Yao et al., 2022) and Toolformer (Schick et al., 2023) showed that interleaving reasoning traces with actions and API calls can substantially extend the capabilities of LLMs. Frameworks such as LangChain and LlamaIndex provide composable abstractions (chains, graphs) for orchestrating multi-step LLM applications with tools and external data sources. Across these lines of work,

the workflow structure—how many steps exist, how they are ordered, which tools each can call, and how information is passed—is largely hand-designed and manually maintained.

**Prompt optimization.** Prompt engineering and optimization have become central to improving LLM systems, and a large body of work now studies how to update prompts automatically using data and feedback. For single-call prompts, methods such as ProTeGi (Pryzant et al., 2023), OPRO (Yang et al., 2024), and PromptBreeder (Fernando et al., 2023) iteratively generate and edit candidate instructions using textual feedback or evolutionary search, treating the LLM itself as an optimizer over discrete prompt strings. DSPy (Khattab et al., 2024) introduces a programming model in which LLM pipelines are parameterized modules in a text transformation graph. MIPRO (Opsahl-Ong et al., 2024) builds on this view and directly targets multi-stage

"language model programs," introducing meta-optimization strategies. Works such as Trace (Cheng et al., 2024), GEPA (Agrawal et al., 2025), and TextGrad (Yuksekgonul et al., 2025) push beyond scalar losses, showing that execution traces and natural-language feedback can be treated as generalized gradients. FLOWBOT shares with these approaches the use of textual feedback to optimize prompts, but differs in that it couples this inner prompt optimization with an outer loop that edits the workflow sketch itself, rather than assuming the program structure is fixed.

**Automated design of workflows and agentic systems.** Most closely related to FLOWBOT are approaches that attempt to *automatically* design the structure of LLM workflows or agentic systems, rather than only tuning their prompts. AFlow (Zhang et al., 2025) models agentic workflows directly as code-represented graphs and applies Monte Carlo Tree Search over workflow edits to optimize such graphs against task-specific metrics. ADAS (Hu et al., 2024) proposes a broader research agenda of Automated Design of Agentic Systems, where a meta-agent searches over the space of agent programs to invent novel building blocks, control flows, and tool-use patterns. Recent fully-automated frameworks such as AutoAgent (Tang et al., 2025) similarly aim to construct and adapt agents and multi-agent workflows from natural-language descriptions, relying on iterative code generation and self-play to synthesize and refine agent graphs. Compared with these approaches, FLOWBOT adopts a simpler search space—feedforward chains of LLM calls—but couples workflow search and module learning tightly: the outer loop edits the workflow sketch with textual gradients, while the inner loop re-optimizes module behavior via layer-wise textual backpropagation.

## 6. Conclusion

We introduced FLOWBOT, a simple bilevel optimization framework that jointly induces the structure and behavior of LLM workflows. By treating workflows as modular stacks of LLM calls and performing layer-wise textual backpropagation in an inner loop, while using an outer loop to refine the workflow sketch itself, FLOWBOT brings ideas from neural architecture search into the space of LLM systems. Across multiple tasks, our experiments show that FLOWBOT discovers sensible workflow decompositions and achieves performance competitive with strong baselines.

## Impact Statement

This paper presents work whose goal is to advance the field of Machine Learning. There are many potential societal consequences of our work, none which we feel must be specifically highlighted here.

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

# A. Meta LLM Prompts

This appendix provides the complete system and user prompts used in our optimization pipeline. Table 11 summarizes the notation for each optimizer prompt.

| Notation | Role | Output |
|---|---|---|
| $\theta_{\text{grad-workflow}}$ | Workflow structure gradient | $g_{\mathcal{W}}$ |
| $\theta_{\text{grad-loss}}$ | Final step gradient | $g_K$ from $\mathcal{L}(x_K, y)$ |
| $\theta_{\text{grad-backprop}}$ | Chain rule gradient | $g_k$ from $g_{k+1}$ |
| $\theta_{\text{optim-workflow}}$ | Workflow optimization (outer loop) | $\mathcal{W}'$ |
| $\theta_{\text{optim-call}}$ | Prompt optimization (inner loop) | $\theta_i'$ |
| – | Executor initialization | New $\theta_i$ |

*Table 11.* Optimizer prompts and their roles in the bilevel optimization pipeline.

## A.1. Textual Gradient Computation

These prompts implement textual backpropagation, computing natural language gradients that describe how each component's output should change to improve task performance.

### A.1.1. $\theta_{\text{GRAD-WORKFLOW}}$: WORKFLOW STRUCTURE GRADIENT

This prompt computes the workflow-level gradient $g_{\mathcal{W}}$, which identifies structural issues in the workflow itself rather than individual step outputs. Given the complete execution trace $\{x_1, \ldots, x_K\}$, evaluation result, and available tools, it analyzes whether the workflow structure (step ordering, missing steps, tool usage) should be modified. This drives the outer loop of bilevel optimization.

---

**System Prompt:** $\theta_{\text{grad-workflow}}$

```
You are computing a textual gradient for the WORKFLOW STRUCTURE.

Your task: Analyze the execution trace and evaluation result to identify
what STRUCTURAL changes would improve performance.
Focus on: step ordering, missing steps, redundant steps, role allocation,
tool usage.

Output valid JSON only.
```

---

**User Prompt:** $\theta_{\text{grad-workflow}}$

```
## Sample Information
**Question**: {question}
**Ground Truth**: {ground_truth}

## Evaluation Result
{evaluation_result}

## Evaluation Metrics
{metrics_info}

## Available Tools
{available_tools}

## Current Workflow Structure
{workflow_structure}

## Execution Trace
```

---

```
{execution_trace}

---

## Your Task

**Step 1: Understand the task.** What type of question is this? What would
be the ideal approach to solve it?

**Step 2: Analyze the execution.** Looking at the execution trace and
evaluation result, what went wrong? Was it a structural issue or just an
execution quality issue?

**Step 3: Propose structural improvements.** What changes to the workflow
structure would help solve this type of question better?

Consider:
- Is the step ordering optimal for this task?
- Are there missing steps that should be added? (e.g., verification,
  intermediate processing, tool usage)
- Are there redundant steps that could be merged or removed?
- Should some steps use tools instead of pure LLM reasoning, or vice versa?
- Is the responsibility allocation between steps appropriate?

Respond in JSON:
```json
{{
    "reasoning": "Your analysis: What type of task is this? What's the
     ideal workflow? What structural issues exist in the current workflow?",
    "text_gradient": "Your specific structural recommendations. Be concrete
     about: what steps to add/remove/reorder/modify, and why."
}}
```
```

### A.1.2. $\theta_{\text{GRAD-LOSS}}$: FINAL STEP GRADIENT

This prompt computes the initial textual gradient $g_K$ for the final step of the workflow. Given the workflow's output $x_K$, ground truth $y$, and evaluation result $\mathcal{L}(x_K, y)$, it generates natural language feedback describing what should change to reduce the loss. This is analogous to computing $\partial\mathcal{L}/\partial x_K$ in neural network backpropagation.

**System Prompt: $\theta_{\text{grad-loss}}$**

```
You are computing a textual gradient for the FINAL step of an AI workflow.

Your task: Given the final output and evaluation feedback, explain what
should change.
Think like backpropagation: trace the error from the loss back to this
step's output.

Output valid JSON only.
```

**User Prompt: $\theta_{\text{grad-loss}}$**

```
## Evaluation Context
**Question**: {question}
**Ground Truth**: {ground_truth}
```

```
## Evaluation Metrics
{metrics_info}

## Final Step Execution
**Step {step_id}**: {step_description}
**Executor**: {executor_name}
**Tools**: {executor_tools}
**Input received**:
{step_input}

**Output produced**:
{step_output}

## Evaluation Result
{evaluation_result}

---

## Your Task
Compute the textual gradient for this final step.

Analyze both **what works well** and **what needs improvement**:

**Working Well (preserve these in the executor prompt):**
- What aspects of the output are correct or well-structured?
- What reasoning approaches should be maintained?

**Needs Improvement (if any):**
- What went wrong (diagnosis)
- How the output should differ (prescription)
- Specific changes needed (concrete suggestions)

Write a comprehensive natural language analysis that includes:
1. What the executor is doing well (to preserve)
2. What needs to change to improve metrics (if any)

Respond in JSON:
```json
{{
    "text_gradient": "Your comprehensive textual gradient here. Include
     both what works well (to preserve) and what needs improvement
     (with specific suggestions)."
}}
```
```

### A.1.3. $\theta_{\text{GRAD-BACKPROP}}$: CHAIN RULE GRADIENT

This prompt propagates textual gradients backward through intermediate steps using the chain rule. Given step $k$'s input $x_{k-1}$, output $x_k$, the next step's output $x_{k+1}$, and the downstream gradient $g_{k+1}$, it computes $g_k$ describing how $x_k$ should change to improve downstream performance. This implements the textual analog of $g_k = f(x_k, x_{k+1}, g_{k+1})$.

**System Prompt: $\theta_{\text{grad-backprop}}$**

```
You are computing a textual gradient using the CHAIN RULE.

Your task: Given this step's output, the next step's output, and the next
step's gradient, propagate the gradient backward to this step.

Think: "How should THIS step's output change so that the NEXT step would
perform better?"
```

```
Output valid JSON only.
```

**User Prompt:** $\theta_{\text{grad-backprop}}$

```
## Chain Rule Context
**Question**: {question}

## This Step (Step {step_id})
**Description**: {step_description}
**Executor**: {executor_name}
**Tools**: {executor_tools}
**Input received (x_{prev_step})**:
{step_input}

**Output produced (x_{step_id})**:
{step_output}

## Next Step (Step {next_step_id})
**Description**: {next_step_description}
**Output (x_{next_step_id})**:
{next_step_output}

## Gradient from Next Step (g_{next_step_id})
{next_gradient}

---

## Your Task (Chain Rule)
Propagate the gradient: g_{step_id} = ∇LLM(x_{step_id}, x_{next_step_id},
g_{next_step_id})

Analyze both **what works well** and **what needs improvement**:

**Working Well (preserve these):**
- What aspects of this step's output correctly support the next step?
- What information or formatting is helpful?

**Needs Improvement (if any):**
- What information is missing that the next step needs?
- What should be added/changed to help downstream steps?

Write a comprehensive natural language analysis that includes:
1. What this step is doing well for downstream (to preserve)
2. What changes would improve downstream performance (if any)

Respond in JSON:
```json
{{
    "text_gradient": "Your comprehensive textual gradient here. Include
     both what works well (to preserve) and what needs improvement
     (with specific suggestions for downstream benefit)."
}}
```
```

## A.2. Bilevel Optimization

These prompts apply the computed textual gradients to update the workflow. The outer loop optimizes the workflow structure $\mathcal{W}$, while the inner loop optimizes individual prompts $\theta_i$.

A.2.1. $\theta_{\text{OPTIM-WORKFLOW}}$: WORKFLOW SKETCH OPTIMIZATION (OUTER LOOP)

This prompt implements the outer loop of bilevel optimization. Given the current workflow structure $\mathcal{W}$, existing executor specifications, and aggregated workflow gradients $\{g_{\mathcal{W}}^{(1)}, \ldots, g_{\mathcal{W}}^{(B)}\}$, it decides whether to modify the workflow structure and produces an updated execution plan $\mathcal{W}'$. This enables adding, removing, or reordering steps based on task requirements.

---

**System Prompt: $\theta_{\text{optim-workflow}}$**

```
You are designing an optimal workflow structure based on feedback from
multiple samples.

Your role:
- Analyze aggregated gradients from multiple samples to find common patterns
- Design a workflow that can solve ALL samples effectively
- Think about the ideal workflow structure for this type of task

Output valid JSON only.
```

---

**User Prompt: $\theta_{\text{optim-workflow}}$**

```
## Evaluation Metrics
{metrics_info}

## Available Tools
{available_tools}

---

## Current Workflow Structure (W)
{current_workflow}

## Current Agents
{current_agents}

---

## Aggregated Workflow Gradients from {num_samples} samples

{aggregated_gradients}

---

## Your Task: Design Optimal Workflow

**Step 1: Find Common Patterns**
Look at all {num_samples} samples above. What structural issues appear
repeatedly? What types of mistakes are being made?

**Step 2: Think About Ideal Workflow**
Given these samples and available tools, what would be the ideal workflow
structure to solve ALL of them effectively? Think from scratch - don't be
constrained by the current structure.

**Step 3: Compare and Decide**
Compare your ideal workflow with the current structure. If they're
significantly different, propose changes. If the current structure is
already optimal, keep it.

Respond in JSON:
```json
```

```
{{
    "reasoning": "1) Common patterns in gradients: ... 2) Ideal workflow
     for this task type: ... 3) Comparison with current: ...",
    "should_update": true or false,
    "updated_execution_plan": [
        {{
            "step_id": 1,
            "description": "step description",
            "tools": ["tool_name"] or [],
            "executor_type": "reuse",
            "executor_name": "exact_name_from_current_agents",
            "generation_guideline": ""
        }},
        {{
            "step_id": 2,
            "description": "New step description",
            "tools": [],
            "executor_type": "new",
            "executor_name": "new_unique_name_v1",
            "generation_guideline": "Detailed guidance for creating this
             executor's prompt. What should it do, how should it format
             output, etc."
        }}
    ]
}}
```

**Decision Criteria:**
- `should_update: true` -> Your ideal workflow differs from current. Propose
  the improved structure.
- `should_update: false` -> Current workflow is already optimal for these
  samples. Leave `updated_execution_plan` empty.

**Tools assignment:**
- `tools: ["tool_name"]` -> Step uses tools (ToolExecutor will be created
  automatically)
- `tools: []` -> Step uses pure LLM reasoning (LLMExecutor will be created
  automatically)

**executor_type rules:**
- `"reuse"`: Use an existing executor. `executor_name` MUST exactly match
  a name from Current Agents. Leave `generation_guideline` empty.
- `"new"`: Create a new executor. `executor_name` MUST be UNIQUE. Provide
  detailed `generation_guideline` for the executor generator.

{output_format_instruction}
```

### A.2.2. $\theta_{\text{OPTIM-CALL}}$: PROMPT OPTIMIZATION (INNER LOOP)

This prompt implements the inner loop of bilevel optimization. Given an executor's current prompt $\theta_i$ and aggregated textual gradients $\{g_i^{(1)}, \ldots, g_i^{(B)}\}$ from a batch of samples, it produces an updated prompt $\theta_i'$ that addresses the identified issues while preserving working patterns. This is analogous to gradient descent: $\theta \leftarrow \theta - \eta \nabla_\theta \mathcal{L}$.

**System Prompt:** $\theta_{\text{optim-call}}$

```
You are applying Textual Gradient Descent (TGD) to update an AI agent's
prompt.

Your role:
```

```
- Take the current prompt and aggregated gradient feedback
- Apply the gradient to produce an improved prompt
- Preserve aspects that work while fixing identified issues
- Ensure the updated prompt generalizes across all feedback samples

Think like gradient descent: move the prompt in the direction that reduces
loss.

Output valid JSON only.
```

**User Prompt:** $\theta_{\text{optim-call}}$

```
## Current Agent
**Name**: {executor_name}
**Type**: {executor_type}
**Tools Available**: {executor_tools}

## Current Prompt (θ)
```
{current_prompt}
```

---

## Aggregated Gradients (∇L) from {num_samples} samples

{aggregated_gradients}

---

## Your Task: Apply TGD

Update the prompt based on the gradients:
θ_new = TGD(θ, ∇L)

Rules:
1. Address ALL identified issues from the gradients
2. Preserve patterns that weren't criticized
3. Make the prompt generalizable (not overfitting to specific samples)
4. Structure the prompt with these sections:
   - Task Description: What the agent does and its purpose
   - Input Format: Expected input structure
   - Approach: Step-by-step strategy and rules for processing
   - Output Format: Expected output structure

Respond in JSON:
```json
{{
    "updated_prompt": "The complete updated prompt text",
    "changes_made": ["Change 1", "Change 2", ...],
    "reasoning": "Why these changes address the gradients"
}}
```
```

## A.3. Workflow Initialization

When the outer loop adds new steps to the workflow (with executor_type: "new"), this module initializes prompts for the new agents. Given the step description, available tools, and generation guidelines, it creates an initial prompt that will be further refined through subsequent inner loop iterations. This is analogous to initializing new layer parameters when

a neural network architecture changes.

---

**System Prompt: Agent Initialization**

```
You are an expert at initializing new agents in LLM workflows.

Your role:
- Create effective prompts that guide agents to complete their assigned
  steps
- Design prompts that generalize across similar inputs
- Configure tool usage when applicable

Similar to initializing layer parameters in neural networks.

Output valid JSON only.
```

---

**User Prompt: Agent Initialization**

```
## Step Information

- **Step ID**: {step_id}
- **Step Description**: {step_description}
- **Tools**: {tools}

## Generation Guideline

{generation_guideline}

## Sample Questions This Executor Should Handle

{questions}

## Your Task

Create an ExecutorSpec with a clear, effective prompt that:
1. Follows the generation guideline above
2. Can handle all types of questions like the samples above (be general,
   not question-specific)
3. Uses the specified tools appropriately (if any)

## Output Format

```json
{{
  "name": "DescriptiveExecutorName",
  "type": "ToolExecutor" or "LLMExecutor",
  "description": "What this executor does",
  "prompt": "The system prompt for this executor..."
}}
```

## Prompt Writing Guidelines

- Be specific about the task and expected output
- If using tools, explain when and how to use each tool
- Handle edge cases (e.g., no results found, ambiguous queries)
```

---

## B. Bilevel Optimization Example

We provide a detailed example of Bilevel Optimization from our HotpotQA experiments. This example illustrates both workflow sketch optimization and prompt optimization on a failing case.

### B.1. Task Information

- **Question**: "The 1905-1904 college basketball team lead by G.W. Leach represented a university founded in what year?"

- **Ground Truth**: 1846 (Bucknell University)

- **Prediction**: 1865

- **Metric**: Exact Match (em)

### B.2. Workflow Sketch Optimization

The workflow sketch $\mathcal{W}$ was updated from 4 steps (1 tool) to 6 steps (2 tools). Two new steps were added: an LLM step ($a_2'$) for entity disambiguation and a tool step ($a_3'$) for targeted attribute retrieval.

| Step | Description | Type | Change |
|------|-------------|------|--------|
| *Before:* $\mathcal{W} = [a_1, a_2, a_3, a_4]$ *(1 tool)* | | | |
| 1 | Wikipedia search | Tool | – |
| 2 | Answer extraction | LLM | – |
| 3 | Verification | LLM | – |
| 4 | Final answer | LLM | – |
| *After:* $\mathcal{W}' = [a_1', a_2', a_3', a_4', a_5', a_6']$ *(2 tools)* | | | |
| 1 | Wikipedia search | Tool | reuse |
| 2 | Entity disambiguation | LLM | **new** |
| 3 | Targeted attribute retrieval | Tool | **new** |
| 4 | Answer extraction | LLM | reuse |
| 5 | Verification | LLM | reuse |
| 6 | Final answer | LLM | reuse |

*Table 12.* Workflow sketch before and after optimization.

### B.2.1. WORKFLOW GRADIENT ($g_{\mathcal{W}}$)

---

**Workflow Gradient** $g_{\mathcal{W}}$

Modify the workflow to explicitly separate entity identification and attribute retrieval steps. After the initial broad Wikipedia search for the question context (Step 1), add a dedicated entity disambiguation step that confirms the relevant university by querying specifically for 'G.W. Leach 1904-1905 basketball team university' or similar, possibly with a retrieval-augmented LLM. Then, perform a targeted retrieval step to get founding year information about the identified university (e.g., a separate wikipedia_search_topk call with the university name and keywords like 'founding year'). Next, extract the founding year from these dedicated passages. Follow this with a verification step that cross-checks multiple passages and confidence scores to confirm the final answer. This division reduces ambiguity and error propagation. Additionally, consider merging answer extraction and verification into a single retrieval-augmented step to use external evidence rather than pure LLM reasoning. Allocate tool usage not only to initial retrieval but also to targeted retrieval steps for entity attributes. Finally, ensure that the final answer generation step formats the answer but does not perform reasoning, keeping responsibilities clear and modular.

---

### B.3. Prompt Optimization via Textual Backpropagation

The textual gradients are computed layer-by-layer using the chain rule, propagating from the loss $\mathcal{L}(x_K, y)$ backward through each step to update prompts $\theta_1, \ldots, \theta_K$.

### B.3.1. $g_6$: FINAL ANSWER → LOSS

---
**Gradient $g_6$: Final Answer**

The executor is correctly following the instruction to provide a concise, exact answer without any additional commentary or explanation, which aligns perfectly with the prompt requirements for the final step. This focused approach is beneficial as it avoids ambiguity and ensures the output format is clean and directly comparable to the ground truth. However, the key issue lies in the accuracy of the answer itself: the executor output '1865' does not match the ground truth year '1846'. This indicates a failure in retrieving or verifying the correct founding year of the university associated with the 1904-1905 college basketball team led by G.W. Leach. To improve, the executor needs to ensure the input answer provided to this step is accurate by cross-referencing reliable data sources or prior steps that confirm the founding year as 1846. Specifically, the executor's input should be corrected upstream, or if this step has any responsibility to verify or correct the answer, it should implement a validation mechanism. In summary, preserve the executor's strict adherence to output formatting and brevity, but improve the accuracy of the final answer by enhancing data retrieval or validation processes before this final step.

---

### B.3.2. $g_5$: VERIFICATION

---
**Gradient $g_5$: Verification**

This step does an excellent job of clearly and thoroughly verifying and ranking the candidate answer, presenting a detailed logical reasoning process that identifies the University of Kentucky as the relevant institution and establishes 1865 as its founding year. The explicit stepwise breakdown, the consideration of historical context, and the ranking of candidate answers provide strong, well-structured support for downstream steps. The formatting is clear and consistent, making it easy for subsequent steps to extract and use the final verified answer. These aspects should be preserved as they ensure clarity, traceability, and well-supported conclusions. However, the main issue is the inaccuracy of the foundational data: the step concludes that 1865 is the correct founding year, but the ground truth is 1846. To improve downstream performance, this step needs to incorporate more robust verification mechanisms, such as cross-checking multiple authoritative sources or explicitly handling discrepancies between candidate answers and common knowledge. It should also consider whether alternative interpretations or predecessor institutions correspond to 1846, or whether the initial premise about the university identity or coach association might be incorrect. Including a mechanism to flag uncertainty or to retain multiple plausible candidates with their confidence scores could help prevent propagating incorrect definitive answers. In sum, preserve the clear reasoning and structured presentation, but improve the accuracy and validation process to ensure the final answer aligns with the correct ground truth.

---

### B.3.3. $g_4$: ANSWER EXTRACTION

---
**Gradient $g_4$: Answer Extraction**

This step correctly extracts and outputs a concise candidate answer string, '1846', which is clearly formatted and readily usable for downstream verification. The simplicity and directness of the output support the next step's ability to take the candidate answer and perform a detailed validation and ranking process. Preserving this clarity and exactness is important for traceability and ease of interpretation by subsequent steps. However, to improve downstream performance, this step should incorporate additional verification signals or confidence indicators alongside the candidate answer. For example, it could include a brief note or metadata referencing the source(s) from which '1846' was extracted or an initial confidence score, which would help the next step assess the reliability of the candidate answer more effectively. Moreover, given that the next step identifies '1846' as inconsistent with authoritative historical data, this step could be enhanced by performing preliminary cross-checks or highlighting uncertainty if conflicting data is detected. Including multiple candidate answers with associated confidence levels, or explicitly flagging potential discrepancies at this stage, would prevent propagating a potentially inaccurate definitive answer and provide richer information for downstream reasoning. In summary, retain the clear, exact answer extraction as is, but augment the output with verification context, confidence metrics, and/or candidate alternatives to enable more robust validation and improve final answer accuracy.

---

B.3.4. $g_3$: TARGETED ATTRIBUTE RETRIEVAL

---

**Gradient $g_3$: Targeted Attribute Retrieval**

This step is performing well by retrieving and presenting a focused set of Wikipedia passages that are directly relevant to the entity 'Bucknell University' and its founding year, which is the key attribute of interest. The output is well-structured, showing multiple passages with clear references to Bucknell University and including the precise founding year '1846' in Passage 5. This clarity and specificity provide the next step with rich, targeted evidence to extract the exact answer string. The formatting is consistent and easy to parse, supporting straightforward extraction of answer candidates. However, to enhance downstream performance, this step could improve by incorporating preliminary verification or confidence signals about the attribute information present in the retrieved passages. For example, it could highlight or annotate the passage(s) containing the founding year explicitly (e.g., Passage 5) to signal higher relevance or confidence. Furthermore, the step could provide multiple candidate founding years if any ambiguity or conflicting data were detected, along with confidence scores or provenance metadata referencing the source passages. Such augmentation would enable the next step to perform more robust validation and ranking of candidate answers, preventing premature commitment to a single definitive answer when uncertainty exists. Including summary notes about potential conflicts or corroborating evidence across passages would also be beneficial. In summary, preserve the current approach of retrieving targeted, relevant passages with clear references to the founding year and maintain the clean, organized formatting. Augment the output by adding verification context, confidence indicators, explicit provenance annotations, and, if applicable, alternative candidate answers. These additions would improve traceability, support richer downstream validation, and ultimately enhance the accuracy and reliability of the final answer extraction.

---

B.3.5. $g_2$: ENTITY DISAMBIGUATION

---

**Gradient $g_2$: Entity Disambiguation**

This step is doing well by clearly identifying and verifying the primary entity relevant to the question, 'Bucknell University,' with explicit reasoning that links G.W. Leach and the 1905-1906 basketball team to that university. The output preserves the entity name consistently and provides a concise, focused explanation that supports the next step's targeted search. This clarity and precision in entity disambiguation and confirmation should definitely be preserved to ensure that subsequent retrieval focuses on the correct subject. To improve downstream performance, this step could be enhanced by incorporating preliminary confidence indicators or metadata about the certainty of the identified entity, especially if multiple candidate entities or ambiguities exist at this stage. For example, including a confidence score or a brief note on whether alternative entities were considered and ruled out would aid the next step in prioritizing searches and interpreting retrieval results. Additionally, providing a succinct summary of the key attribute of interest (e.g., 'founding year') alongside the entity in this step's output could enable the next step to better tailor its search queries or filtering criteria. Finally, if any potential ambiguities or known alternative names for the entity exist, mentioning those explicitly would help downstream retrieval be more comprehensive and robust. In summary, maintain the strong, clear entity verification and consistent formatting while augmenting the output with confidence signals, explicit mention of the attribute(s) of interest, and any relevant provenance or alternative entity considerations. These additions will enhance traceability, enable more focused and confident retrieval in the next step, and support accurate and reliable final answer extraction.

---

B.3.6. $g_1$: WIKIPEDIA SEARCH

---

**Gradient $g_1$: Wikipedia Search**

This step effectively retrieves relevant Wikipedia passages containing key information about the 1905-1906 college basketball team led by G.W. Leach, which supports the next step's entity disambiguation. The output includes multiple passages, among them a clear identification of the Bucknell Bison team with G.W. Leach as captain. This breadth of information and the inclusion of specific, detailed passages help the next step confidently select the primary entity. The formatting of the passages, with clear labels and relevant details, is also useful for downstream processing. However, to better support subsequent steps, this output could be improved by explicitly highlighting or annotating the most relevant passage(s) or sentence(s) that directly link G.W. Leach to Bucknell University. Adding a brief summary or extracted fact snippet such as 'G.W. Leach was captain of the 1905–06 Bucknell Bison men's basketball team' would make the primary entity linkage more salient and easier to parse for later steps. Furthermore, including metadata or preliminary confidence indicators regarding the retrieved passages or candidate entities would enhance traceability and downstream reasoning. For example, a confidence score or a note indicating that Bucknell University is the strongest candidate based on available data would help the entity disambiguation step prioritize and verify entities more effectively. Finally, considering potential alternative entities or ambiguities at this stage—if any were identified—would improve robustness by preparing the next step to handle multiple candidates or conflicting data. Overall, preserving the detailed, well-structured retrieval output while adding explicit relevance annotations, confidence signals, and a concise fact extraction will significantly improve downstream entity verification and answer extraction accuracy.

---

## B.4. Error Analysis

This example reveals an interesting failure mode:

1. **Step 3** (Targeted Retrieval): Correctly retrieved passages containing "Bucknell was founded in 1846"

2. **Step 4** (Extraction): Correctly extracted "1846" from the passages

3. **Step 5** (Verification): **LLM Hallucination** occurred
   - The LLM incorrectly stated: "G.W. Leach is historically known as a coach of the Kentucky Wildcats"
   - This caused the model to reject the correct answer (1846) and substitute 1865 (Kentucky's founding year)

Here, while retrieval-augmented workflows provided correct evidence, downstream LLM reasoning steps were incorrect.. The textual gradients correctly identified this issue and suggested incorporating "more robust verification mechanisms" and "cross-checking multiple authoritative sources."

## B.5. Prompt Update Example

Based on the textual gradients computed above, the verification step's prompt was updated to address the identified hallucination issue. We show the complete prompt before and after the update.

### B.5.1. BEFORE UPDATE ($\theta_5$)

---
**Verification Step: Original Prompt**

You are given multiple candidate answers to a question, each supported by one or more passages and associated confidence indicators. Your task is to carefully verify the correctness and reliability of each candidate answer by cross-checking the supporting passages for consistency, factual accuracy, and strength of evidence. Consider any confidence indicators provided as part of your assessment. If some answers conflict, analyze the evidence to determine which answer is most trustworthy. If no answer is sufficiently supported, indicate that no reliable answer could be determined. Finally, rank the candidate answers from most to least reliable and provide a clear explanation for your ranking and the final selected answer.

---

### B.5.2. AFTER UPDATE ($\theta_5'$)

---
**Verification Step: Updated Prompt**

You are given multiple candidate answers to a question, each supported by one or more passages and associated confidence indicators. Your task is to carefully verify the correctness and reliability of each candidate answer by cross-checking the supporting passages for consistency, factual accuracy, and strength of evidence. Consider any confidence indicators provided as part of your assessment. If some answers conflict, analyze the evidence to determine which answer is most trustworthy. **To improve accuracy, explicitly check for discrepancies among candidate answers and, where possible, cross-reference with common knowledge or authoritative sources. When discrepancies or uncertainties arise, flag them clearly and retain multiple plausible candidates along with their confidence scores rather than selecting a definitive answer prematurely.** If no answer is sufficiently supported or uncertainty remains high, indicate that no reliable answer could be determined. Finally, rank the candidate answers from most to least reliable and provide a clear explanation for your ranking and the final selected answer.

---

