# OpenReview forum: "FlowBot: Inducing LLM Workflows with Bilevel Optimization and Textual Gradients"
_ICML.cc/2026/Conference — ICML 2026 regular_

### Official Review · Reviewer_TpAy · 2026-03-10

**Soundness:** 2
**Presentation:** 3
**Significance:** 3
**Originality:** 3
**Overall Recommendation:** 3
**Confidence:** 3

**Summary:**

This paper presents WIBOT (Workflow Induction through Bilevel Optimization and Textual Gradients), a framework for automatically inducing and optimizing LLM workflows without relying on hand-crafted pipelines. The authors formulate workflow induction as a bilevel optimization problem where an outer loop optimizes the high-level workflow sketch (structure and composition of LLM calls) while an inner loop optimizes individual prompts within a fixed sketch. Both levels leverage "textual gradients"—natural language feedback generated by an LLM based on inputs, outputs, and losses. The inner loop implements layer-wise textual backpropagation, treating the workflow as a modular stack and propagating gradients backward from the final loss to update each component while conditioning on downstream behavior. The outer loop uses workflow-level textual gradients to refine the sketch itself, adding, removing, or reordering steps. Drawing inspiration from neural architecture search, this approach couples structure search with module learning. Evaluated across ten diverse benchmarks spanning multi-hop QA, instruction following, verification, privacy-conscious delegation, mathematical reasoning, and code generation, WIBOT demonstrates competitive performance against strong baselines including GEPA for prompt optimization and AFlow for automated workflow generation, with particularly strong gains on retrieval-intensive tasks.

**Compliance With Llm Reviewing Policy:**

Affirmed.

**Key Questions For Authors:**

How does the computational cost of WIBOT compare to AFlow and other baselines in terms of total LLM API calls and wall-clock time, and have you conducted experiments to determine the scaling behavior as the number of training examples or workflow steps increases?

What is the sensitivity of the method to the quality of the meta-LLM used for gradient generation—have you experimented with smaller or less capable models for computing textual gradients, and does performance degrade gracefully or collapse?

The current formulation restricts workflows to linear chains; have you explored extensions to handle branching, loops, or conditional execution, and what technical challenges would need to be addressed to support such control flow structures?

In the ablation study, removing textual backpropagation causes performance degradation, but how much of this is due to the loss of modularity versus simply having fewer optimization steps—would a non-modular approach with equivalent computational budget perform similarly?

The outer loop can add, remove, or reorder steps, but what prevents it from generating workflows with redundant or circular structures, and are there explicit constraints or regularization terms to encourage parsimony?

Finally, the initialization procedure starts from a minimal workflow and builds up complexity—have you explored alternative initialization strategies (such as starting from a complex workflow and pruning) or bidirectional search, and how sensitive are the final results to the initial structure?

**Limitations:**

See Weaknesses and Key Questions For Authors.

**Strengths And Weaknesses:**

**Strengths**

The authors present the context of LLM workflow automation as a critical challenge in scaling complex AI systems, and their bilevel formulation with textual gradients offers a conceptually elegant synthesis of ideas from neural architecture search and recent work on prompt optimization. The layer-wise backpropagation of textual gradients is a particularly compelling technical contribution, enabling modular credit assignment in ways that end-to-end workflow search methods like AFlow cannot achieve. The detailed case study in Appendix A provides excellent qualitative insight into how textual gradients identify specific failure modes (such as hallucination in verification steps) and guide targeted prompt updates without disrupting functioning upstream components. The experimental breadth across ten benchmarks covering diverse task types strengthens the generalizability claims, and the controlled comparison using the same base model (GPT-4.1 mini) for both WIBOT and reimplemented AFlow ensures fair evaluation. The ablation studies convincingly demonstrate that both the bilevel structure and the layer-wise gradient propagation are necessary for optimal performance.

**Weaknesses**
However, several weaknesses temper the enthusiasm for this work. First, the restriction to linear chain workflows significantly limits expressiveness compared to graph-based or branching workflows that many real-world applications require; while the authors acknowledge this as future work, the current formulation cannot handle conditional execution, loops, or parallel processing that would be essential for general agentic systems.

Second, the computational cost of the approach is substantial and not adequately characterized—each optimization step requires multiple LLM calls for gradient computation and prompt updates, and the paper provides no analysis of total API costs or comparison with the query budgets of baseline methods.

Third, the evaluation primarily uses relatively small batch sizes (5 samples) and limited training epochs (one epoch with two bilevel loops), raising questions about scalability to larger datasets and whether the method would continue improving or plateau with more data.

Fourth, the reliance on textual gradients introduces potential brittleness: the quality of optimization depends heavily on the meta-LLM's ability to correctly diagnose errors and prescribe fixes, yet there is no analysis of gradient quality or failure modes where the meta-LLM might generate misleading gradients.

Finally, while the paper positions WIBOT against neural architecture search, it lacks the systematic search exploration that characterizes NAS—there is no discussion of how the outer loop explores the space of possible workflows or whether it might get stuck in local optima.

---

> ### Author Rebuttal · Authors · 2026-03-31
>
> We thank the reviewer for a thoughtful review! We first respond to the questions.
>
> **Q1. How does the computational cost of WIBOT compare to AFlow and other baselines in terms of total LLM API calls...**
>
> We provide a cost comparison against AFlow, which is the method that is most comparable to ours (since it also induces workflow structure). Across all benchmarks We find that our approach uses ~80% fewer LLM calls and is ~50% cheaper.
>
> | Benchmark | AFlow API Calls | AFlow Cost | WIBOT API Calls |  WIBOT Cost |
> | --------- | --------------: | -------------: | --------------: | ----------: |
> | HotpotQA  |         491,000 |        $227.21 |          61,835 |      $76.39 |
> | DROP      |         163,000 |         $30.39 |          23,395 |      $12.62 |
> | HumanEval |          13,365 |          $2.79 |           2,208 |       $3.27 |
> | MBPP      |          52,460 |         $18.94 |           7,262 |      $10.34 |
> | GSM8K     |          63,360 |         $45.82 |          18,102 |       $9.41 |
> | MATH      |          20,230 |         $27.32 |          31,843 |      $53.36 |
> | **Total** |     **803,415** |    **$352.46** |     **144,645** | **$165.39** |
>
> **Q2. What is the sensitivity of the method to the quality of the meta-LLM**
>
> Great question! We ran our approach on HotpotQA2 with a weaker meta LLM (gpt-oss-20B), and finds that the performance does degrade, but does so gracefully.
>
> | Dataset | Meta-LLM | Execution LLM | Score  |
> |---|---|---|---:|
> | hotpotqa2 (AFlow) | gpt-4.1-mini | gpt-4.1-mini | 0.8112 |
> | |  gpt-oss-20b  | gpt-4.1-mini | 0.7558 |
>
>
> **Q3. The current formulation restricts workflows to linear chains; have you explored extensions to handle branching, loops, or conditional execution?**
>
> We find that a simple addition to the prompt allows it to induce workflows with loops and chains. Here's an example on HotpotQA, where it learns to loop in step 2.
>
> - **Step 0 (LLM):** Pre-analysis
> - **Step 1 (Tool):** Search + validate
> - **Step 2 (Loop / LLM):** Evidence verify
>   - If `need_more_retrieval = true`, return to **Step 1** (maximum 2 iterations)
>   - If `need_more_retrieval = false`, proceed to **Step 3**
> - **Step 3 (Conditional / LLM):** Answer generation gate
>   - If `true`, proceed to **Step 4 (LLM):** Final answer
>   - If `false`, proceed to **Step 5 (LLM):** Info not found handler
>
> The performance of this model for HotpotQA is 0.7566, and the full workflow + prompt is given here: https://docs.google.com/document/d/e/2PACX-1vQUCIcWoTqszQsp4_x9nzjbzlQuknxLm0swSym2iNr_1F-F3FcV9vlPMjj9qpxsPqfAXDD5iaDh2d_W/pub
>
> In response to Q4 of Reviewer wr1G we induce and even more complex input-dependent workflows, where we branch based the input.
>
> **Q4. In the ablation study, removing textual backpropagation causes performance degradation, but how much of this is due to the loss of modularity versus simply having fewer optimization steps**
>
> Sorry for the confusion: we actually apply the same number of optimization steps for the backpropagation-removed baseline, so our comparison here is fair.
>
> **Q5. The outer loop can add, remove, or reorder steps, but what prevents it from generating workflows with redundant or circular structures, and are there explicit constraints or regularization terms to encourage parsimony?**
>
> We do not have such explicit constraints. In preliminary work we tried encouraging parsimony with prompts like "Make it as simple as possible, but no simpler", but found that this did not change performance much.
>
> **Q6. Finally, the initialization procedure starts from a minimal workflow and builds up complexity—have you explored alternative initialization strategies**
>
> We were primarily focused on inducing workflows from scratch, as in AFlow/ADAS. But this is a great suggestion, and we will explore this for future work.
>
> We also respond to some weaknesses:
>
> **Third, the evaluation primarily uses relatively small batch sizes (5 samples) and limited training epochs (one epoch with two bilevel loops), raising questions about scalability to larger datasets and whether the method would continue improving or plateau with more data.**
>
> This is a great point, but we generally find this to be the case for prompt/workflow optimization style methods (see, e.g., Figure 1 of GEPA). This could also be seen as a feature, insofar as it enables data efficiency :)
>
> **the reliance on textual gradients introduces potential brittleness: the quality of optimization depends heavily on the meta-LLM's ability to correctly diagnose errors and prescribe fixes, yet there is no analysis of gradient quality or failure modes where the meta-LLM might generate misleading gradients.**
>
> We do show some qualitative examples of gradients in the paper (e.g., Figure 3). We agree that such an analysis would be interesting, but doing so systematically is difficult. If the reviewer has any suggestions, we'd be happy to take them (and do them during the discussion period!)

---

> > ### Author Rebuttal · Reviewer_TpAy · 2026-04-03
> >
> > Thanks to the author for their detailed response to the review, especially regarding the cost comparison with other benchmark solutions and the details of the extended linear analysis experiments. I have read the author's rebuttal and other reviewers' comments. However, it seems the method may still be somewhat sensitive to LLM quality, as I have only looked at one dataset and a relatively large model so far. Furthermore, regarding the training and evaluation mentioned, as scaling inevitably increases the size of the dataset or model, prompt/workflow optimization may become less suitable and, to some extent, may be replaced by a stronger base model. Therefore, based on these issues, I am currently inclined to maintain the current score.

---

> > > ### Author Response · Authors · 2026-04-05
> > >
> > > Thanks for engaging in the discussion process! We understand your concerns regarding the limited experiments regarding the sensitivity of WIBOT to the different models. We performed additional experiments with a wider range of models, on both HotpotQA-1 and HotPotQA-2, to address your concerns.
> > >
> > > **HotPotQA-1 (GEPA)**
> > > | Executor | Meta-LLM | No Opt | WIBOT (prompt) | WIBOT (all) |
> > > |---|---|---:|---:|---:|
> > > | Qwen3.5-4B | Qwen3.5-122B-A10B | 0.417 | 0.513 ± 0.008 | 0.681 ± 0.014 |
> > > | Qwen3.5-9B | Qwen3.5-122B-A10B | 0.497 | 0.615 ± 0.015 | 0.737 ± 0.010 |
> > > | Qwen3.5-27B | Qwen3.5-122B-A10B | 0.618 | 0.677 ± 0.007 | 0.779 ± 0.010 |
> > > | Qwen3.5-122B-A10B | Qwen3.5-122B-A10B | 0.670 | 0.667 ± 0.009 | 0.746 ± 0.011 |
> > > | Qwen3.5-397B-A17B | Qwen3.5-122B-A10B | 0.682 | 0.732 ± 0.007 | 0.740 ± 0.012 |
> > >
> > > **HotPotQA2 (AFlow)**
> > > | Executor | Meta-LLM | No Opt | WIBOT (prompt) | WIBOT (all) |
> > > |---|---|---:|---:|---:|
> > > | Qwen3.5-4B | Qwen3.5-122B-A10B | 0.608 | 0.615 ± 0.007 | 0.761 ± 0.004 |
> > > | Qwen3.5-9B | Qwen3.5-122B-A10B | 0.677 | 0.700 ± 0.003 | 0.772 ± 0.003 |
> > > | Qwen3.5-27B | Qwen3.5-122B-A10B | 0.774 | 0.775 ± 0.009 | 0.800 ± 0.003 |
> > > | Qwen3.5-122B-A10B | Qwen3.5-122B-A10B | 0.771 | 0.730 ± 0.003 | 0.817 ± 0.004 |
> > > | Qwen3.5-397B-A17B | Qwen3.5-122B-A10B | 0.784 | 0.816 ± 0.005 | 0.806 ± 0.004 |
> > >
> > > Here, "No Opt" is just using the LLM without any prompt optimization, "WIBOT (prompt)" just runs the prompt optimization loop of WIBOT, and "WIBOT (all)" uses both prompt optimization and workflow induction.
> > >
> > > We observe several trends:
> > > - **Even with stronger base models, WIBOT (all) improves upon standard prompting**. We hope this addresses your concern about *"as scaling inevitably increases the size of the dataset or model, prompt/workflow optimization may become less suitable and, to some extent, may be replaced by a stronger base model."*
> > > - **Strong Meta-LLMs enable significant improvements upon prompt-only optimization.** We observe that with Qwen3.5-122B as the Meta-LLM, this can enable significant improvements with smaller Executor LLMs. The improvements diminish with a larger executor LLM (i.e., Qwen3.5-397B), which we think makes sense, given that (in some sense) the Meta-LLM's job is more difficult.
> > > - Since the actual deployment will be with the executor LLMs, only requiring powerful LLMs for the Meta-LLM component is in our opinion quite practical---we could use an expensive Meta-LLM at "training time" for workflow/prompt induction, and then deploy the cheap execeutor LLMs.
> > >
> > > We thank the reviewer for these interesting additional suggestions, and will add the results to the paper.
> > >
> > > Please let us know if this addresses your concerns to an extent---and if it does, please consider updating your score.
> > >
> > > Thank you!

---

### Official Review · Reviewer_wr1G · 2026-03-11

**Soundness:** 3
**Presentation:** 2
**Significance:** 3
**Originality:** 3
**Overall Recommendation:** 3
**Confidence:** 3

**Summary:**

This paper proposes WIBOT, a method for automatically building LLM workflows instead of relying on humans to hand-design the pipeline. The key idea is to cast workflow induction as a bilevel optimization problem: an outer loop edits the overall workflow structure, while an inner loop optimizes the prompt of each step using textual gradients. A central design is layer-wise textual backpropagation, where downstream errors are translated into natural-language feedback and propagated backward to improve upstream modules. The paper evaluates WIBOT on ten benchmarks spanning multi-hop QA, instruction following, privacy delegation, math, and code generation, and reports competitive or better performance than strong baselines, while also showing that the induced workflows are often sensible and similar to human-designed decompositions.

**Compliance With Llm Reviewing Policy:**

Affirmed.

**Final Justification:**

I think the paper is at the boraderline of acceptance overall.

**Key Questions For Authors:**

(1) Can the authors provide a clearer cost–performance analysis of WIBOT, especially compared with simpler baselines?

(2) Maybe give an ablation study disentangle when the gains come from workflow structure search versus prompt optimization?

(3) Any evidence to showcase that the learned workflows are robust and not overly task- or split-specific?

(4) Can authors evaluate WIBOT on workflows that require richer control flow than a serialized chain?

**Limitations:**

yes

**Strengths And Weaknesses:**

Strength: Empirically, the paper is fairly strong: it evaluates on a broad set of tasks, beats or matches strong automated baselines on most datasets, and includes ablations showing that both the bilevel setup and the layer-wise textual backpropagation matter. The qualitative analyses are another plus: the paper shows induced workflows that look reasonable and gives a concrete error-analysis example where the backward textual gradients correctly identify the step that hallucinated and revise only that prompt.

Weakness: The clearest limitation is that the workflow representation is still restricted to a linear chain of LLM calls, so richer agent behaviors like branching, looping, and conditional control flow are not truly modeled, even if the authors argue they could be serialized. Another limitation is optimization cost: both the outer workflow edits and inner prompt updates require many extra LLM calls, so the method may be expensive and may not be worthwhile when simple workflows already work well. The paper also admits that on some tasks the method converges early and stops benefiting from more data, which suggests the learning procedure may still be somewhat rigid. Natural langauge gradients may also cascading errors and error propagation, which can make the method less stable in real-world.

---

> ### Author Rebuttal · Authors · 2026-03-31
>
> Thank you for your thoughtful review!
>
> We first respond to the weakness:
>
> **The clearest limitation is that the workflow representation is still restricted to a linear chain of LLM calls...**
>
> We initially went with the serial approach since many tasks within the workflow induction literature are simple enough to address with a chain. Based on your suggestion, we modified the prompt slightly to allow for richer workflows, and find that it is able to induce loops. Here's an example on HotpotQA, where it learns to loop in step 2.
>
> - **Step 0 (LLM):** Pre-analysis
> - **Step 1 (Tool):** Search + validate
> - **Step 2 (Loop / LLM):** Evidence verify
>   - If `need_more_retrieval = true`, return to **Step 1** (maximum 2 iterations)
>   - If `need_more_retrieval = false`, proceed to **Step 3**
> - **Step 3 (Conditional / LLM):** Answer generation gate
>   - If `true`, proceed to **Step 4 (LLM):** Final answer
>   - If `false`, proceed to **Step 5 (LLM):** Info not found handler
>
> The performance of this model for HotpotQA is 0.7566, and the full workflow + prompt is given here: https://docs.google.com/document/d/e/2PACX-1vQUCIcWoTqszQsp4_x9nzjbzlQuknxLm0swSym2iNr_1F-F3FcV9vlPMjj9qpxsPqfAXDD5iaDh2d_W/pub
>
> (In response to Q4 we induce even more complex workflows).
>
> **Q1. Can the authors provide a clearer cost–performance analysis...**
>
> We provide a cost comparison against AFlow, which is the method that is most comparable to ours (since it also induces workflow structure). Across all benchmarks We find that our approach uses ~80% fewer LLM calls and is ~50% cheaper.
>
> | Benchmark | AFlow API Calls | AFlow Cost | WIBOT API Calls |  WIBOT Cost |
> | --------- | --------------: | -------------: | --------------: | ----------: |
> | HotpotQA  |         491,000 |        $227.21 |          61,835 |      $76.39 |
> | DROP      |         163,000 |         $30.39 |          23,395 |      $12.62 |
> | HumanEval |          13,365 |          $2.79 |           2,208 |       $3.27 |
> | MBPP      |          52,460 |         $18.94 |           7,262 |      $10.34 |
> | GSM8K     |          63,360 |         $45.82 |          18,102 |       $9.41 |
> | MATH      |          20,230 |         $27.32 |          31,843 |      $53.36 |
> | **Total** |     **803,415** |    **$352.46** |     **144,645** | **$165.39** |
>
> **Q2.  Maybe give an ablation study disentangle when the gains come from workflow structure search versus prompt optimization?**
>
> Here are the results of our approach on HotpotQA where we just do prompt optimization
>
> | Dataset | Model |  Score  |
> |---|---|---:|
> | hotpotqa1 (GEPA) | WIBOT | 0.7280 |
> | | Prompt optimization only  | 0.6007 |
> | hotpotqa2 (AFlow) | WIBOT | 0.8019 |
> | | Prompt optimization only  | 0.7206 |
>
>
> As shown above, we find that the bilevel optimization step is crucial.
>
> **Q3. Any evidence to showcase that the learned workflows are robust and not overly task- or split-specific?**
>
> We think that task-specific workflows are not a bad thing, given that our aim was to induce a workflow for a given task. However, as we show in our response to Q4, we find that we can have one "meta" workflow that can address different tasks.
>
> **Q4. Can authors evaluate WIBOT on workflows that require richer control flow than a serialized chain?**
>
> As a testbed for whether WIBOT can optimize richer control flows, we attempted to see whether WIBOT could induce a single workflow for multiple benchmarks. Here we use Qwen3.5-379B as it was more cost effective and also more capable.
>
> | AFlow Tasks | Separate Workflows | One Workflow |
> |---|---:|---:|
> | HotpotQA2 | 0.8112 | 0.7920 |
> | DROP | 0.9214 | 0.9200 |
> | HumanEval | 0.9740 | 0.8440 |
> | MBPP | 0.7889 | 0.6360 |
> | GSM8K | 0.9727 | 0.9700 |
> | MATH | 0.8025 | 0.6670 |
> | Avg. | 0.8785 | 0.8048 |
>
> (Some of the runs are not finished yet but we commit to providing them during the discussion period).
>
> WIBOT can induce conditional workflows that make sense. For example, in the AFlow case the first step decides whether the task is, and then branches to different steps (see below).
>
> - **Step 1 [multi_conditional]** — classify `task_type`
>   - **"READING_COMPREHENSION"** → Step 2 → Step 3 *(loop)* → Step 4
>     Evidence extraction → Answer + verification loop → QA output
>   - **"MATH_PROBLEM"** → Step 5 → Step 6 *(loop)* → Step 7
>     Problem analysis → Solve + verification loop → Math output
>   - **"CODE_GENERATION"** → Step 8 → Step 9 → Step 10 *(loop)* → Step 11
>     Requirements analysis → Code generation → Verification loop → Code output
>   - **"GENERAL_CONVERSATION"** → Step 11
>     General output
>
> The induced workflow can be found here: https://docs.google.com/document/d/e/2PACX-1vTG_U7RlLg98eWQR3dnSNmLEuV2PUsUt_3gjoLAqwDHSnKPaBUDFZAI-IrqFhptqSuy-uMO6DfXxn0t/pub
>
> We will add this experiment (and more qualitative examples) to the next iteration of the paper.

---

> > ### Author Rebuttal · Reviewer_wr1G · 2026-04-04
> >
> > I updated my ratings, thank you for the explanations.

---

> > > ### Author Response · Authors · 2026-04-04
> > >
> > > Hi there! Thank you very much for engaging in the discussion process, as well as agreeing to update the ratings. However, it seems like you may have forgotten to actually update the rating, as it has not been changed from the original 3 rating.
> > >
> > > Would you mind updating the ratings on the official review, in case you have forgotten to do so? Thanks!

---

### Official Review · Reviewer_kyrn · 2026-03-15

**Soundness:** 2
**Presentation:** 3
**Significance:** 3
**Originality:** 2
**Overall Recommendation:** 4
**Confidence:** 3

**Summary:**

This paper proposes WIBOT (Workflow Induction through Bilevel Optimization and Textual gradients), a framework for automatically discovering and optimizing LLM workflows, i.e., structured sequences of LLM calls augmented with different instructions and tool access. The key idea is to formulate workflow induction as a bilevel optimization problem, drawing an analogy to neural architecture search (NAS). The outer loop searches over the high-level workflow structure (the "sketch"), determining how many LLM calls to include, their ordering, and their functional roles (e.g., retrieval, verification, reasoning). The inner loop, given a fixed workflow structure, optimizes the prompt of each individual LLM call through a layer-by-layer textual backpropagation procedure. Specifically, textual gradients, i.e., natural language feedback generated by an LLM, are computed at the final layer from the loss, then propagated backward through intermediate layers via a chain-rule-like mechanism, and finally used by an optimizer LLM to update each layer's prompt. Both levels of optimization are driven by textual gradients: the inner loop uses per-layer gradients for prompt refinement, while the outer loop uses workflow-level gradients to propose structural modifications (adding, removing, or reordering steps). The authors evaluate WIBOT on ten benchmarks spanning multi-hop QA, mathematical reasoning, code generation, instruction following, and privacy-conscious delegation, using GPT-4.1 mini as the executor LLM. They report competitive or superior performance compared to prompt optimization baselines (GEPA, TextGrad, MIPROv2, Trace) and automatic workflow generation methods (AFlow, ADAS), and provide ablations showing that both the bilevel formulation and layer-wise backpropagation contribute meaningfully to performance.

**Compliance With Llm Reviewing Policy:**

Affirmed.

**Key Questions For Authors:**

1. **Comparison with Deep Language Networks.** The layer-by-layer textual backpropagation appears very similar to [1]. A detailed conceptual and empirical comparison is essential.

2. **Computational cost.** Please report total API calls, monetary cost, and wall-clock time vs. GEPA and AFlow on representative benchmarks.

3. **Gradient quality measurement.** How often do textual gradients correctly identify the responsible layer, and does quality degrade with chain depth?

[1] Sordoni et al., "Deep Language Networks: Joint Prompt Training of Stacked LLMs using Variational Inference," arXiv 2024.

**Limitations:**

The authors acknowledge several limitations in Section 4: the restriction to chain-structured workflows, the computational overhead of bilevel optimization, and the observation that in some cases the model converges quickly without leveraging all training data. These are honest disclosures. However, additional limitations that should be discussed include: (1) the complete dependence on textual gradient quality with no formal guarantees; (2) the reliance on a specific commercial API (GPT-4.1 mini) with no evaluation of robustness across models; (3) potential overfitting of workflow structure to the training distribution (the paper picks the best workflow on validation, but does not discuss generalization of the workflow itself to distribution shifts); and (4) the lack of any mechanism to handle tasks where the optimal workflow structure is input-dependent (i.e., different inputs may benefit from different workflows). The paper also does not discuss broader impacts or failure modes beyond the hallucination example.

**Strengths And Weaknesses:**

### Strengths

1. **Elegant bilevel formulation.** The NAS-inspired framing, where the outer loop handles workflow structure search and the inner loop handles prompt optimization, is conceptually clean and fills a genuine gap in the LLM workflow literature.

2. **Layer-wise textual backpropagation.** Extending textual gradients to multi-layer workflows via chain-rule-like backward passes is non-trivial; Section 3.5 and Figure 3 effectively demonstrate error-source identification and corrective feedback propagation.

3. **Broad experimental coverage.** 10 benchmarks spanning QA, math, code, instruction following, and privacy delegation, with both prompt-optimization (Table 1) and workflow-generation (Table 2) baselines.

### Weaknesses

1. **Superficial NAS analogy.** Unlike classical NAS with continuous relaxations and well-defined gradients, textual gradients are noisy and non-differentiable with no convergence guarantees, so the formal properties of bilevel optimization do not transfer.

2. **No gradient quality analysis.** The entire method depends on textual gradient reliability, yet no systematic measurement exists: error attribution accuracy, cross-layer degradation, and failure rates are all unexamined.

3. **Questionable comparison fairness.** Different base models (GPT-4.1 mini vs. Claude-3.5-Sonnet + GPT-4o mini), different starting points (scratch vs. human-designed), and no cost reporting make efficiency and performance comparisons unreliable.

---

> ### Author Rebuttal · Authors · 2026-03-31
>
> We thank the reviewer for a thoughtful review.
>
> **Q1. Comparison with Deep Language Networks.**
>
> Thank you for highlighting the connection. While the works are related in that both use language feedback to optimize prompts, our work is differentiated by the fact that we also induce workflows as part of our algorithm. DLN is therefore more of a baseline for TextGrad and GEPA, since these works both fix the workflow structure and only does prompt optimization. Nonetheless, this is an important work to discuss, and we will do so in the next iteration of the paper.
>
> **Q2. Computational cost.**
>
> We provide a cost comparison against AFlow, which is the method that is most comparable to ours (since it also induces workflow structure). Across all benchmarks we find that our approach uses ~80% fewer LLM calls and is ~50% cheaper.
>
> | Benchmark | AFlow API Calls | AFlow Cost | WIBOT API Calls |  WIBOT Cost |
> | --------- | --------------: | -------------: | --------------: | ----------: |
> | HotpotQA  |         491,000 |        $227.21 |          61,835 |      $76.39 |
> | DROP      |         163,000 |         $30.39 |          23,395 |      $12.62 |
> | HumanEval |          13,365 |          $2.79 |           2,208 |       $3.27 |
> | MBPP      |          52,460 |         $18.94 |           7,262 |      $10.34 |
> | GSM8K     |          63,360 |         $45.82 |          18,102 |       $9.41 |
> | MATH      |          20,230 |         $27.32 |          31,843 |      $53.36 |
> | **Total** |     **803,415** |    **$352.46** |     **144,645** | **$165.39** |
>
> **Q3. Gradient quality measurement.**
>
> This is a great question. Qualitatively, we find that the gradient model is often able to localize the error to the right "layer", as shown in Figure 3 of the submission. Doing a systematic study of this is difficult. If the reviewer has a suggestion on how we might be able to do this more systematically, we are open!
>
> We also address some of the weaknesses/limitations
>
> **Different base models (GPT-4.1 mini vs. Claude-3.5-Sonnet + GPT-4o mini)...**
>
> We don't believe this is a concern since our main competing baseines (AFlow/GEPA) use the same GPT-4.1 mini model. (We reran the AFlow baseline using GPT-4.1 to ensure a fair comparison). The starting points between WIBOT and AFlow are the same. GEPA starts from a human-designed workflow, so it is arguably more advantaged than WIBOT, and hence our comparable performance GEPA only strengthens our contribtuion.
>
> **Restriction to chain-structured workflows**
>
> We initially went with the serial approach since many tasks within the workflow induction literature are simple enough to address with a chain. During the rebuttal we modified the prompt slightly to allow for richer workflows, and find that it is able to induce loops. Here's an example on HotpotQA, where it learns to loop in step 2.
>
> - **Step 0 (LLM):** Pre-analysis
> - **Step 1 (Tool):** Search + validate
> - **Step 2 (Loop / LLM):** Evidence verify
>   - If `need_more_retrieval = true`, return to **Step 1** (maximum 2 iterations)
>   - If `need_more_retrieval = false`, proceed to **Step 3**
> - **Step 3 (Conditional / LLM):** Answer generation gate
>   - If `true`, proceed to **Step 4 (LLM):** Final answer
>   - If `false`, proceed to **Step 5 (LLM):** Info not found handler
>
> The performance of this model for HotpotQA is 0.7566, and the full workflow + prompt is given here: https://docs.google.com/document/d/e/2PACX-1vQUCIcWoTqszQsp4_x9nzjbzlQuknxLm0swSym2iNr_1F-F3FcV9vlPMjj9qpxsPqfAXDD5iaDh2d_W/pub
>
> This highlights the promise of our approach in extending to richer workflow structures (which we also explore in inducing input-dependent workflow structures, discussed below).
>
> **(1) the complete dependence on textual gradient quality...**
>
> This is indeed true, though it is a limitation of all text-based prompt optimization methods. We will discuss this more thoroughly in the next iteration of the paper.
>
> **(2) the reliance on a specific commercial API (GPT-4.1 mini)...**
>
> We test our approach on open-weight models, and find that it works well, as shown below
> | Dataset | Model |  Score  |
> |---|---|---:|
> | hotpotqa1 (GEPA) | qwen3.5-397b-a17b | 0.7726 |
> | | gpt-oss-120b | 0.6946 |
> | | gpt-oss-20b | 0.5406 |
> | hotpotqa2 (AFlow) | qwen3.5-397b-a17b | 0.8112 |
> | | gpt-oss-120b | 0.8061 |
> | | gpt-oss-20b |  0.7716 |
>
>
> **(3) potential overfitting of workflow structure to the training distribution...**
>
> We find that the induced workflows are generally similar across runs and moreover robust to distribution shifts. For example, changing the validation set results in 1~2\% standard deviation across 5 runs on HotpotQA.
>
> **(4) the lack of any mechanism to handle tasks where the optimal workflow structure is input-dependent**
>
> We attempted to see whether WIBOT could induce a single workflow for multiple benchmarks, i.e., where the structure is input dependent. We find that it is able to do so, as shown in our response to Reviewer w1RG's Q4.

---

> > ### Author Rebuttal · Reviewer_kyrn · 2026-04-06
> >
> > I thank the authors for the thorough rebuttal, particularly the cost comparison with AFlow and the extension to non-linear workflows with loops and conditionals. These additional results strengthen the contribution. As my original score was already a weak accept, I am inclined to maintain my current score.

---

### Decision · Program_Chairs · 2026-04-30

**Decision:**

Accept (regular)

**Comment:**

This paper introduces a framework (WIBOT) for discovering and optimizing LLM workflows.
Their method is based on reformulating workflow designs with a bilevel optimization problem similar to neural architecture search (NAS). The outer loop seachers over a workflow sketch while the inner loop optimizes each prompt using the sketch.
I actually like the idea of hierarchical design of plans and i think is novel. The paper also is quite honest in acknowledging its limitations which I also appreciated. The reviewers raised several concerns and after my reading of the author's rebuttals I think the authors addressed them sufficiently.
Given the novelty in the method and good evaluation, I recommend acceptance.